# Nucleocytoplasmic Trafficking Perturbation Induced by Picornaviruses

**DOI:** 10.3390/v13071210

**Published:** 2021-06-23

**Authors:** Belén Lizcano-Perret, Thomas Michiels

**Affiliations:** de Duve Institute, Université Catholique de Louvain, VIRO B1.74.07, 74, Avenue Hippocrate, 1200 Brussels, Belgium; belen.lizcano@uclouvain.be

**Keywords:** picornavirus, nuclear pore complex, nucleoporins, leader (L) protein, 2A protease, 3C protease, RAN GTPase, karyopherin

## Abstract

Picornaviruses are positive-stranded RNA viruses. Even though replication and translation of their genome take place in the cytoplasm, these viruses evolved different strategies to disturb nucleocytoplasmic trafficking of host proteins and RNA. The major targets of picornavirus are the phenylalanine-glycine (FG)-nucleoporins, which form a mesh in the central channel of the nuclear pore complex through which protein cargos and karyopherins are actively transported in both directions. Interestingly, while enteroviruses use the proteolytic activity of their 2A protein to degrade FG-nucleoporins, cardioviruses act by triggering phosphorylation of these proteins by cellular kinases. By targeting the nuclear pore complex, picornaviruses recruit nuclear proteins to the cytoplasm, where they increase viral genome translation and replication; they affect nuclear translocation of cytoplasmic proteins such as transcription factors that induce innate immune responses and retain host mRNA in the nucleus thereby preventing cell emergency responses and likely making the ribosomal machinery available for translation of viral RNAs.

## 1. Introduction: Nucleocytoplasmic Trafficking of Proteins and Picornavirus Replication

Nucleocytoplasmic transport of proteins and RNA occurs through the nuclear pore complexes (NPC). These complexes, localized in the nuclear envelope, are composed of two main elements: a stationary phase and a soluble phase. Multiple copies of ~30 different nucleoporins (Nups) organized in an eightfold rotational symmetry form the stationary phase [1,2,3]. These Nups can be further divided into scaffold Nups and phenylalanine-glycine (FG)-Nups (Figure 1). Scaffold Nups are important for maintaining the structure and attachment of the complex to the nuclear envelope. FG-Nups contain intrinsically disordered FG-repeat domains that are important for interacting with partners in the soluble phase [4,5,6]. The soluble phase is mainly composed of proteins such as the small GTPase RAN and nuclear transport receptors (karyopherins, which include importins and exportins). The NPC allows the passive diffusion of small proteins (<40 kDa). Larger proteins rely on active transport, which depends on the presence of specific signals on cargo proteins. These signals, nuclear localization signal (NLS) and nuclear export signal (NES) are recognized by nuclear transport receptors, which tug the cargos through the nuclear pore complex. The active transport is depicted in Figure 1. For the import of a cytoplasmic cargo protein, an importin recognizes the NLS of the cargo protein. The importin/cargo complex then goes through the NPC by interacting with the FG-Nups. Once in the nucleoplasm, the importin binds RAN-GTP and this interaction induces the release of the cargo protein (Figure 1 left). For the export of a nuclear cargo protein, an exportin bound to RAN-GTP recognizes the NES of the cargo protein. The RAN-exportin-cargo complex then passes through the pore thanks to the interaction of the exportin with the FG-Nups. Once in the cytoplasm, RAN-GTP is hydrolyzed into RAN-GDP, which triggers the dissociation of the exportin and cargo protein. Finally, RAN-GDP is returned to the nucleus by the nuclear transport factor 2 (NTF2) and is converted into RAN-GTP by the regulator of chromosome condensation 1 (RCC1) (Figure 1 right) [7,8].

Schematic representation of the nuclear pore: FG-Nups that interact with karyopherins: importins and exportins; and scaffold Nups that maintain the structure of the pore and its binding to the nuclear envelope. Left: nuclear import of a cytoplasmic protein. The NLS sequence of the cargo protein is bound by an importin, which drives the cargo through the pore by interacting with FG-Nups. In the nucleus, the binding of RAN-GTP to the importin frees the cargo protein. Right: export of a nuclear protein to the cytoplasm. The NES of the nuclear cargo protein is bound by an exportin complexed with RAN-GTP. The cargo-exportin-RAN-GTP complex translocates to the cytoplasm through the interaction of the exportin with the FG-Nups. Once in the cytoplasm, RAN-GTP is hydrolyzed into RAN-GDP, which induces the dissociation of the cargo-exportin complex. RAN-GDP is brought back to the nucleus by NTF2 and converted into RAN-GTP by the RAN guanine nucleotide exchange factor RCC1 [7,8].

Picornaviruses are small positive-stranded RNA viruses. As of early 2021, the *Picornaviridae* family was composed of 147 species classified in 63 different genera but the family is expanding fast [9]. The most studied picornaviruses include members of the *Enterovirus* (e.g., poliovirus, rhinovirus, enterovirus A71, and coxsackievirus), *Cardiovirus* (e.g., Theiler’s murine encephalomyelitis virus—TMEV, encephalomyocarditis virus—EMCV, and Saffold virus—SAFV) and *Aphthovirus* (e.g., Foot-and-mouth disease virus—FMDV) genera. Their genome is rapidly translated upon delivery in the cytosol. Translation occurs in a cap-independent manner, thanks to the presence of an internal ribosome entry site (IRES) that recruits the translation complex [10,11,12,13]. Viral RNA is translated into a 216 to 277 kDa polyprotein that is mainly processed by the viral protease 3C^pro^. Other proteins contribute to polyprotein processing in a genus-dependent manner. For example, enteroviruses have an additional protease: 2A^pro^, which cleaves the viral polyprotein between VP1 and 2A sequences [14]. Aphthoviruses L^pro^ is the N-terminal extremity of the polyprotein, which cleaves itself out of the polyprotein [14]. For both cardioviruses and aphthoviruses, the 2A protein contains a short sequence that induces a ribosomal skipping “cleavage” between 2A and 2B [15,16].

Nucleocytoplasmic trafficking and its inhibition have been addressed by interesting reviews [7,8,17,18,19]. This review focuses on the case of picornaviruses with some historical perspectives, as picornaviruses were at the forefront of RNA virus studies.

As for most RNA viruses, picornavirus genome translation and replication both take place in the cytoplasm. At first glance, these viruses thus have little need to harness nucleocytoplasmic trafficking of proteins. However, as will be highlighted in this review, picornaviruses from different genera evolved different strategies to target components of the NPC. The purpose of this trafficking perturbation will be discussed.

## 2. Picornaviruses Trigger the Mislocalization of Host Proteins in Infected Cells

The presence of nuclei is not essential for replication of picornaviruses as early studies showed that poliovirus and echovirus could replicate and generate progeny viruses in enucleated cells [20,21] or cytoplasmic extracts [22,23,24]. The titers of newly synthesized viruses obtained from non-nucleated extracts are however much lower than those obtained from whole cells, indicating that nuclei, albeit not essential, do provide factors that contribute to infection efficiency. Such factors were identified later when nuclear proteins, such as Sam68 [25,26], nucleolin [27], La autoantigen [26,28], and polypyrimidine tract binding protein (PTB) [29,30,31] were shown to be recruited to the cytoplasm of infected cells and to interact with the viral genome or with viral proteins to promote viral replication or translation. Relocalization of nuclear proteins to the cytoplasm during picornavirus infection suggested that nucleocytoplasmic trafficking of proteins was impaired by these viruses. In 2000, Belov et al. reported that a 90 kDa protein made of the NLS sequence of SV40 (Simian virus 40) large T antigen and 3xEGFP, thus dependent on active nucleocytoplasmic transport, was mislocalized to the cytoplasm in HeLa cells infected with either poliovirus or coxsackievirus B3 (two enteroviruses). The integrity of the NLS-3xEGFP fusion protein was confirmed by western blotting, thus ruling out that mislocalization resulted from partial degradation of the protein and/or from the loss of the NLS signal [32]. Interestingly, depending on the genera, distinct viral proteins were found to be responsible for nucleocytoplasmic trafficking perturbation, the main triggers being proteases 2A^pro^ and to a lesser extent 3C^pro^ in the case of enteroviruses, and the leader (L) protein in the case of cardioviruses. The individual implication of these proteins will be detailed below (point 2.3).

### 2.1. Which Trafficking Pathways Are Affected?

The next question was whether trafficking of all proteins was affected or whether specific trafficking pathways were targeted by picornaviruses. The impact of picornavirus infection on nucleocytoplasmic trafficking was tested using reporter constructs or host proteins carrying different targeting signals: classical NLS (from SV40 large T antigen) [33], M9-NLS (M9 sequence present in heterogeneous nuclear ribonucleoprotein A1—hnRNPA1) [34], RS-NLS (RS: arginine- and serine-rich domain present in splicing factors of the SR family, from SRSF2 or the entire SC35 protein) [35] and the leucine-rich NES, also called classical NES (from Rev or protein kinase inhibitor (PKI)—both being exportin 1 (CRM1)-dependent) [36]. Cardiovirus L proteins and rhinovirus 2A^pro^ can affect all 4 tested pathways (classical NLS, M9-NLS, RS-NLS, and classical NES) [37,38]. Depending on the rhinovirus genotype, differences in the kinetics and the extent of protein mislocalization were observed. In general, trafficking perturbation by rhinovirus 2A^pro^ occurs with the following selectivity: M9-NLS > RS-NLS > classical NLS > Leucine-rich NES. [38]. Even though rhinovirus 2A^pro^ and cardiovirus L proteins were shown to affect RS-NLS dependent pathways (using reporter constructs), the localization of SC35 (a protein containing an RS-NLS) was not found to be modified in some studies involving rhinovirus [26] or TMEV [31]. An explanation for the discrepancy in the results concerning SC35 could be its partial retention in the nucleus through interaction with other nuclear components. For poliovirus, not all pathways were equally perturbed since the localization of SC35 (protein containing an RS-NLS) was not altered during infection, nor was the localization of a protein fused to the NES of Rev from HIV, but the classical NLS or M9-NLS pathways were found to be disrupted [39].

Alteration of the nucleocytoplasmic transport by picornaviruses also affects RNA export. PolyA+ mRNA was shown to be trapped in the nucleus in cells infected with different picornaviruses [40,41,42].

### 2.2. FG-Nucleoporins as the Main Targets for Nucleocytoplasmic Disturbance

The simultaneous alteration of protein export and import suggested that picornaviruses affect the integrity of the nuclear pore complex. Gustin and Sarnow were the first to document Nups degradation during poliovirus infection. They noticed that two FG-nucleoporins, NUP153 and NUP62, were no longer detectable by western blot after 4.5 h of infection, a timing that correlated with nucleocytoplasmic trafficking perturbation [39]. The same was demonstrated to happen after rhinovirus [26] and coxsackievirus B3 infection [43]. Later, it was shown that another FG-nucleoporin, NUP98, was also degraded during poliovirus infection, but at an earlier time of infection (around 2 h.p.i compared to 4 h.p.i for NUP153 and NUP62). Degradation of NUP98 was not dependent on replication of the viral genome, suggesting that initial translation of the input viral genome was sufficient to induce NUP98 degradation. It was however not sufficient to induce the mislocalization of nucleolin [44], suggesting that higher amounts of viral proteins and other NPC alterations were needed to observe full-blown nucleocytoplasmic perturbation. Krull et al. examined the cleavage of an extended panel of nucleoporins and reported that NUP35, NUP54, NUP58, NUP62, NUP98, NUP153, NUP214, NUP358, POM121, TPR, and, to a lesser extent, NLP1 were cleaved during poliovirus infection. All these nucleoporins but TPR are FG-Nups. Other non-FG-Nups (e.g., NUP160, NUP133, NUP107, NUP96, NUP43) were not degraded during poliovirus infection, clearly indicating a preference for FG-nucleoporins over scaffold nucleoporins [45]. Table 1 summarizes the different nucleoporins targeted by picornaviruses. Park et. al. demonstrated that the FG-rich domains were specifically cut out of the nucleoporins, leaving essentially the scaffolding domain in the NPC [46,47]. The FG-domains of different nucleoporins interact with each other to maintain the permeability barrier [48]. The contribution of the NUP98 FG-domain appears to be particularly important [49]. Picornavirus-induced simultaneous degradation of the FG-domain of several Nups, including NUP98, therefore critically cripples nuclear pore complex function.

### 2.3. Different Picornaviruses Acting with Different Proteins

By treating picornavirus-infected cells with the pan-caspase inhibitor z-VAD-fmk, (which does not affect the activity of 2A^pro^ and 3C^pro^), Gustin and Sarnow showed that the influence of caspases on NUP153 and NUP62 cleavage was negligible in the case of poliovirus and very weak in the case of rhinovirus [26]. Next, it was shown that protease inhibitors such as MPCMK, elastatinal, antipain, or chymostatin, that inhibit 2A^pro^ of poliovirus and rhinovirus, prevented virus-induced mislocalization of reporter-NLS proteins, suggesting a role for 2A^pro^ in nucleocytoplasmic perturbation [50]. Moreover, expression in transfected HeLa cells of 2A^pro^ but not of the proteolytically inactive mutant 2A^pro^ (H20N) leads to the cytoplasmic relocalization of the 3xGFP-NLS fusion [50]. These first results indicated that 2A^pro^ was the main viral protein inducing the nucleocytoplasmic disturbance in poliovirus and rhinovirus infected cells. Next, it was shown, in vitro with purified 2A^pro^ and in vivo in transfected cells, that NUP62, NUP98, and NUP153 were direct targets of rhinovirus 2A^pro^. Interestingly, depending on the genotype of rhinovirus, the 2A^pro^ induced different patterns of cleavage and had different preferences for each nucleoporin [38,40,44,46,47,51,52].

The involvement of 3C^pro^ in Nups degradation was also suggested. Rhinovirus 16 3C^pro^ expression in COS7 cells leads to the degradation of NUP358, NUP214, and NUP153 but not of NUP62 or NUP98 [53,54]. The addition of purified 3C^pro^ on permeabilized cells triggered the diffusion of NLS-GFP out of the nucleus, indicating that nucleocytoplasmic transport was altered by 3C^pro^ [54]. Other attempts to observe NUP98 cleavage by 3C^pro^ from rhinovirus or coxsackievirus in vitro were unsuccessful [46,52]. Only partial cleavage of NUP62 was observed in vitro with 1 µg of 3C^pro^ (incubation of 8 h) [46]. Taken together, these results indicate that in the case of enteroviruses, 2A^pro^ is the main protease targeting the nuclear pore complex but that 3C^pro^ may contribute as well.

Recently, a high throughput degradomic study from Saeed et al., based on the subtiligase technique, identified dozens of proteins that are cleaved during picornavirus infection [55]. As expected, several proteins of the nuclear pore complex were detected in this screen including FG-nucleoporins (NUP214, NUP98, NUP62, NUP58, NUP54), the non-FG nucleoporin (RAE1), and, interestingly, the soluble phase GTPase RAN. NUP98 cleavage was confirmed by western blot for poliovirus, enterovirus A70 and A71, coxsackievirus B3, and rhinovirus A16 [55]. Table 1 provides an overview of nucleoporins shown to be targeted by specific picornaviruses.

**Table 1 viruses-13-01210-t001:** Nucleoporins targeted by picornaviruses.

Nucleoporins	FG-Repeat	*Enterovirus*				*Cardiovirus*	
		Poliovirus	Enterovirus 70 & 71	Coxsackievirus B3	Rhinovirus	EMCV	TMEV
NUP35	+	C [45]	NT	NT	NT	NT	NT
NUP54	+	C [45]	NT	C *	NT	NT	NT
NUP58	+	C [45]	NT	NT	NT	NT	NT
NUP62	+	C [26,39,44,45] (2Apro [40])	NT	C [43]	C [26,53] (2Apro [38,46,47,51], 3Cpro [46])	P [56,57]	P *
NUP98	+	C [44,45,55] (2Apro [40,47])	C [55]	C (2Apro [52,55])	C [53,55] (2Apro [44,47,51])	NT	P [41]
NUP153	+	C [26,39,44,45] (2Apro [40])	NT	C [43]	C [26] (2Apro [38,51], 3Cpro [53,54])	P [56,57]	P *
NUP214	+	C [45]	NT	C *	C (3Cpro [54])	P [56,57]	P *
NUP358	+	C [45]	NT	NT	C (3Cpro [54])	- [56]	NT
NLP1	+	C [45]	NT	NT	NT	NT	NT
POM121	+	C [45]	NT	NT	NT	NT	P *
TPR	-	C [45]	NT	NT	NT	NT	NT
RAE1	-	C *	NT	C *	NT	NT	NT

C: virus-induced cleavage (2A^pro^ or 3C^pro^, when not specified: not known); C *: virus-induced cleavage (Saeed et al.); P: phosphorylation induced by L protein; P *: phosphorylation induced by L protein (Mass spectrometry analysis—our unpublished data); NT: not tested.

Cardioviruses, including TMEV and EMCV, were also shown to disturb nucleocytoplasmic trafficking [31,43]. The 2A protein of cardioviruses is however unrelated to enterovirus 2A^pro^ and lacks protease activity. The first implication of another viral protein in nucleocytoplasmic trafficking perturbation was reported by Delhaye et al., who showed that TMEV-induced mislocalization of the nuclear protein PTB depended on the leader (L) protein [31]. L proteins of cardioviruses are very small proteins of 67–76 amino acids lacking any enzymatic activity [58,59]. Using transfection of EMCV replicons lacking either 2A or L and introduction of in vitro-translated proteins into digitonin-permeabilized cells, Lidsky et al. confirmed that L but not 2A was required for nucleocytoplasmic traffic perturbation [43]. Importantly, in cells infected with EMCV, FG-Nups (NUP62, NUP153, and NUP214) were not degraded but turned out to be hyperphosphorylated [56,57]. Similar observations were made for NUP98 in the case of TMEV [41]. Table 1 lists the nucleoporins that were shown to be targeted by cardioviruses.

Interestingly, introducing GST-L in digitonin-permeabilized cells was sufficient to trigger NPC alteration, demonstrating that the L protein alone (out of a viral infection context) can induce this phenomenon. Moreover, if cells expressing the L protein were treated with Staurosporine (a broad-spectrum kinase inhibitor), nucleoporins were no longer phosphorylated and nucleocytoplasmic transport was restored, suggesting that nucleocytoplasmic traffic disturbance is caused by the phosphorylation of FG-nucleoporins [56] and therefore that the L protein, which has no catalytic activity, likely promoted Nup phosphorylation by a cellular kinase. By treating cells with different protein kinase inhibitors, Porter et al. showed that NPC leakage and FG-nucleoporins phosphorylation were diminished when p38 and/or MEK-ERK pathways were blocked. The size of NUP62 tryptic phosphopeptides indicated that FG-repeat domains are likely the sites of hyperphosphorylation. Also, mostly threonines and serines were found to be phosphorylated, in line with the Ser/Thr kinase activity of ERK, RSK, and p38 [57]. Finally, the L protein of EMCV was shown to be phosphorylated by two cellular kinases: CK2 and SYK. Mutation of phosphorylated L protein residues prevented L-mediated NUP62 hyperphosphorylation, indicating that L protein phosphorylation is a prerequisite to nucleocytoplasmic trafficking inhibition [60]. TMEV and SAFV L can be phosphorylated by AMPK at the level of residues that are not conserved in EMCV L [61]. The impact of these phosphorylations on nucleocytoplasmic transport perturbation has however not yet been tested.

### 2.4. Are Picornaviruses Dismantling the NPC Structure?

FG-Nups cleavage by enteroviruses and phosphorylation by cardioviruses both increase the permeability of the nuclear pore complex. Do these modifications similarly impact the structure of the NPC?

Immunofluorescent microscopy analysis documented the disappearance of FG-Nups such as NUP62 and NUP98 from the nuclear envelope in cells infected with poliovirus [39,45,47], rhinovirus [26,46,47,53], and coxsackievirus [52]. Using antibodies specific for the different domains of the Nups, Krull et al. showed that 2A^pro^-mediated cleavage mostly released the FG-domain of these nucleoporins, leaving the anchoring part of the Nups in the nuclear envelope (see Figure 2B). Scaffold nucleoporins such as NUP107 were still attached to the nuclear envelope, indicating that the nuclear pore complex was not completely dismantled [45]. In the case of cardioviruses, FG-nucleoporins were left in the nuclear envelope and the NPC was thus also not dismantled [62].

Electron microscopy analysis showed neither disappearance of NPCs nor modifications of the overall NPC diameter in cells infected with poliovirus or EMCV [43,50]. Electron-dense material spanning the central channel of the pore disappeared in EMCV-infected cells and was replaced by irregular granules likely corresponding to proteolytic products of FG-Nups in poliovirus-infected cells [62]. Observation of the NPCs after detergent treatment further revealed that structural links formed between Nups or between NPC and the lamina were affected after poliovirus but not EMCV infection. The structural integrity of the NPC was thus more deeply affected after poliovirus infection [62].

### 2.5. RAN and Karyopherins Are among the Targets in the Soluble Phase

Nucleoporins alteration is likely the main cause of nucleocytoplasmic trafficking disturbance by picornaviruses. However, components of the soluble phase: RAN-GTPase and karyopherins were also reported to be targeted by picornaviruses. The first indication that soluble phase components were targeted by picornaviruses was a report by Porter et. al. showing that the L protein of cardioviruses interacts with the small GTPase RAN [42]. The hinge domain of the L protein was identified as the interacting domain of L with RAN, as confirmed by solution structures of the L-RAN complex [63,64,65]. The relationship between L-RAN interaction and nucleoporins phosphorylation is however not yet fully clear. Mutations in the hinge domain of L prevent the interaction with RAN and decrease NUP62 phosphorylation, indicating that phosphorylation of FG-Nups might require L interaction with RAN. However, mutations in other parts of L (such as mutation 4D4A, in the acidic domain) abolished nucleoporin phosphorylation but did not abolish interaction with RAN, suggesting that Nup phosphorylation might be independent of the L-RAN interaction [63]. Using GST pull-down experiments, Ciomperlik et. al. showed that the L protein not only interacted with RAN but also with nuclear transport receptors. Two karyopherins, exportin 1 (also called CRM1) and exportin 2 (also called CAS), were pulled down from HeLa cell lysates, by GST-L. The interaction of L with CRM1 was not dependent on the presence of RAN but size exclusion chromatography suggested the assembly of trimeric L-RAN-CRM1 complexes (or larger combinations). When CRM1 knock-down cells were infected with EMCV, NUP62 phosphorylation was diminished but not totally abrogated, as were the amounts of viral proteins, indicating that CRM1 down-regulation was inhibiting viral replication [66]. In summary, the L protein of cardioviruses likely forms a complex with RAN and CRM1 and induces the phosphorylation of FG-nucleoporins, which leads to permeabilization of the NPC. As proposed by A. Palmenberg [64,66] the L-RAN-CRM1 complex might recruit a kinase such as ERK1/2 or p38 that would be responsible for the nucleoporins’ phosphorylation. Alternatively, L-RAN-CRM1 complex formation and FG-Nups phosphorylation may be two independent ways used by cardioviruses to cripple the nucleocytoplasmic traffic machinery.

Aphthoviruses and enteroviruses also target NPC soluble phase components. In the case of the foot-and-mouth disease virus (*Aphthovirus*), protease 3C^pro^ was shown to trigger the degradation of karyopherin subunit α1 (KPNA1) in a proteasome and caspase-independent manner [67]. Surprisingly, enterovirus A71 was reported to induce the degradation of the same protein but in a 2A^pro^- and 3C^pro^-independent but caspase-dependent manner [68]. Enterovirus A71 also affects karyopherin subunit-alpha 2 (KPNA2). Unexpectedly, infection by enterovirus A71 triggered the transcriptional upregulation of this karyopherin [69].

## 3. Conclusions and Discussion

Picornaviruses belonging to different genera evolved different strategies to target the nuclear pore complex and to perturb nucleocytoplasmic trafficking of proteins and RNA. The main targets of these viruses are the phenylalanine-glycine-rich domains of FG-Nups. However, some picornaviruses also act on components of the soluble phase of the NPC by targeting RAN and/or the nuclear transport receptors (Figure 2A).

A likely purpose of nucleocytoplasmic disturbance is the recruitment, to the cytosol, of nuclear proteins that promote viral replication and/or translation (Figure 3). Interestingly, cytoplasmic relocalization of nuclear proteins not always depends on NPC perturbation as illustrated by the case of the La autoantigen. In poliovirus-infected cells, La autoantigen has been shown to migrate into the cytoplasm and to bind the 5′ non-coding region of the poliovirus genome, thereby stimulating IRES-dependent translation [28,70,71]. Shiroki et al. showed that the NLS sequence of the La autoantigen was cleaved out by the viral protease 3C^pro^, thereby inducing its cytoplasmic localization (Figure 2A and Figure 3) [72].

Another likely purpose of NPC targeting by picornaviruses is the inhibition of innate immunity signaling. Most antiviral innate immunity pathways depend on the nuclear translocation of transcription factors that are activated by cytoplasmic kinases in response to viral infection. Preventing the access of such transcription factors to the nucleus thus prevents transcriptional upregulation of genes coding for innate immunity mediators such as interferon. One such example is the interferon regulatory factor 3 (IRF3) that displayed aberrant localization and phosphorylation during cardiovirus infection and failed to induce interferon-α/β gene transcription [31,41,73]. Other transcription factors implicated in the interferon signaling pathway are the STAT proteins. Karyopherin subunit α1 was shown to be degraded in a 3C^pro^-dependent manner in FMDV-infected cells and a caspase-dependent manner after enterovirus A71 infection. This karyopherin is responsible for the translocation of phosphorylated STAT1 into the nucleus. So, by triggering KPNA1 degradation, both FMDV and enterovirus A71 prevent the translocation of STAT1/2 into the nucleus, thus antagonizing the transcriptional upregulation of interferon-stimulated genes [67,68] (Figure 3). In contrast, enterovirus A71-triggered the upregulation of karyopherin 2α gene transcription. This upregulation likely contributes as a proinflammatory signal, as this karyopherin allows the nuclear translocation of proteins such as p65, IRF1, TP53, or ERK1/2 [69].

At last, nucleocytoplasmic transport disruption during picornavirus infection was reported to block host polyA+ mRNA export. By doing so, the virus may prevent the translation of cell mRNAs coding for antiviral proteins. Moreover, this would leave ribosomes directly available for viral mRNA translation, as these mRNAs are generated in the cytoplasm and therefore, do not undergo nuclear export (Figure 3).

Picornaviruses are not the only pathogens that interact with NPC components. Viruses that replicate in the nucleus, such as DNA viruses and RNA viruses from the *Orthomyxoviridae* family (e.g., Influenza virus), need to get their genome in the nucleus. Therefore, these viruses’ proteins and genomes interact with NPC components and use nucleocytoplasmic transport to get their genome into the nucleus [74]. Nonetheless, other RNA viruses, such as picornaviruses, that replicate in the cytoplasm also target the NPC. Striking examples are Dengue and Zika viruses of the *Flaviviridae* family since they encode a protease called NS3, which also targets FG-nucleoporins such as NUP153 and NUP98 to trigger mislocalization of cellular components between the nucleus and the cytoplasm [75]. Another timely example is the coronavirus SARS-CoV-2: ORF6 encoded by this virus interacts and misplaces NUP98 and RAE1 [76]. This induces a bidirectional perturbation of the nucleocytoplasmic traffic, a retention of mRNA in the nucleus [77], and a blockade of STAT1/2 dimer translocation into the nuclei resulting in interferon signaling inhibition [78]. These effects are very similar to the ones induced by picornaviruses.

Interestingly, non-virus pathogens were also shown to target the nucleocytoplasmic traffic machinery. Bacteria such as *Salmonella*, *Coxiella,* and *Orientia* counteract innate immune defenses and notably NFκB activation by targeting exportins, importins, or RAN [79,80,81]. Thus, pathogens as different as picornaviruses and bacteria evolved diverse manners to target the NPC and to perturb the nucleocytoplasmic traffic, probably in part with the common goal to escape innate immunity.

## Figures and Tables

**Figure 1 viruses-13-01210-f001:**
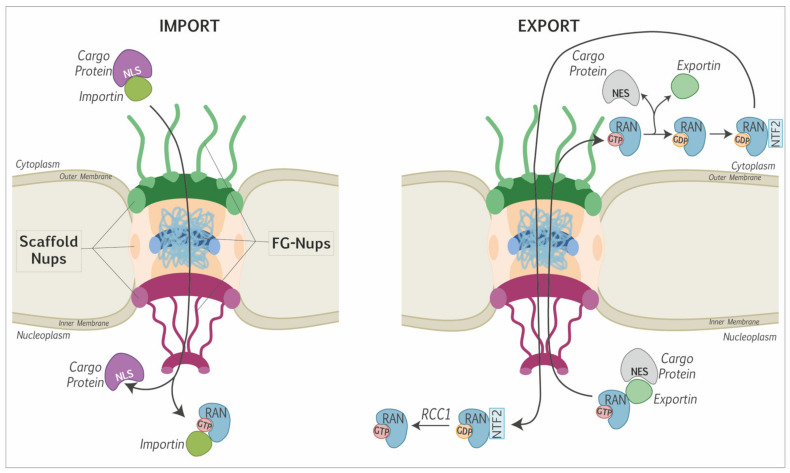
The nuclear pore complex and nucleocytoplasmic transport of proteins.

**Figure 2 viruses-13-01210-f002:**
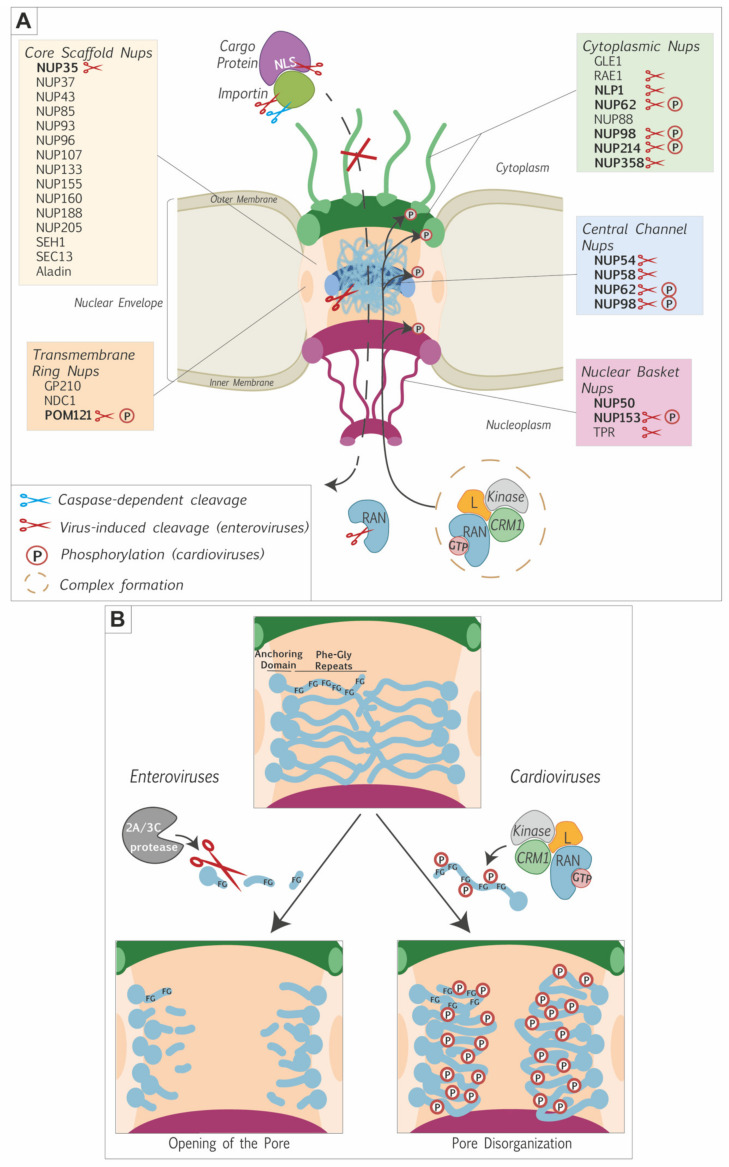
Picornavirus-induced alteration of the nuclear pore complex. (**A**) Illustration of the NPC showing alterations triggered by picornaviruses. The proteins forming the different nuclear pore components and some soluble phase proteins are indicated. FG-nucleoporins are in bold characters. Alterations triggered by picornaviruses are depicted as follows. Blue scissors: caspase-dependent degradation (e.g., importins); Red scissors: virus-dependent cleavage; P: phosphorylation. The model proposed to account for NUP phosphorylation by cardioviruses is illustrated: a complex formed between the L protein, RAN and exportins would recruit kinases (e.g., ERK1/2 and p38) to the NPC, where these kinases would induce the phosphorylation of FG-nucleoporins. (**B**) Zoom on FG-nucleoporins molecular modifications by picornaviruses. Enteroviruses use their 2A^pro^ and 3C^pro^ to cleave the FG domains, thereby inducing an opening of the pore. Cardioviruses use their L protein to trigger the phosphorylation of the FG-rich domains of FG-nucleoporins, thereby inducing disorganization of the pore.

**Figure 3 viruses-13-01210-f003:**
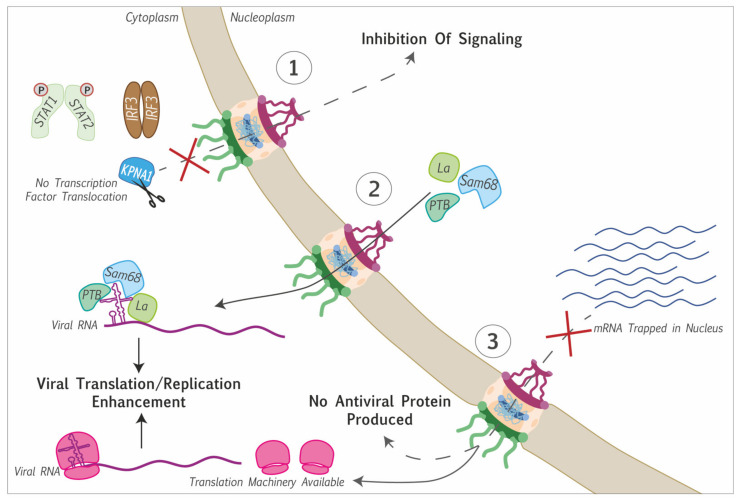
Consequences of protein and RNA trafficking perturbation induced by Picornaviruses. (1) Cytoplasmic retention of transcription factors (STAT1/2, IRF3) thereby inhibiting transcriptional induction of cellular genes: e.g., karyopherin subunit α1 cleavage prevents the translocation of STAT1/2 into the nucleus. (2) Nuclear proteins are delocalized to the cytoplasm, where they interact with the viral genome to promote viral genome translation or replication. (3) Blocking of mRNA export, preventing the translation of antiviral proteins, and making the translation machinery available for viral mRNA translation.

## Data Availability

Not applicable.

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
