# Peer review of "Nucleocytoplasmic Trafficking Perturbation Induced by Picornaviruses"

_viruses, 2021, doi:10.3390/v13071210_

Round 1

Reviewer 1 Report

Nucleocytoplasmic trafficking perturbation induced by picornaviruses

By B  Lizcano Perret and T Michiels* (*Corresponding author)

Submitted to Viruses (Editorial No viruses – 1246815)

General Comments

This review summarizes data on the interaction of different proteins of viruses belonging to different genera of the Picornaviridae family with cellular proteins, in particular proteins located in the nuclear membrane and the nuclear pore complex. Regrettably, the presentation is not very transparent. There is some confusion in the citation of some of the references. In particular, assessments of molecular mechanisms of how picornaviruses disturb cellular nucleocytoplasmic interactions are missing.

Specific Comments

Line

14        FG-nucleoproteins… Spell out at first mentioning.

25ff     The introduction should contain relevant references for the different compounds/genes introduced. Fig. 1 is insufficiently explained, and the involvement of picornavirus proteins is not obvious.

57ff     It is suggested to establish a Table in which the proteins of different picornavirus genera involved in interaction with the nuclear pore complex/nuclear functions are listed including relevant refs.

107      Provide relevant refs.

122      Table 1 is rather non-discriminatory and could be omitted.

190      Table 2 is not very illuminating. In particular, a review of possible molecular mechanisms of picornavirus protein-nuclear pore complex interactions is missing in the following section (Lines 199ff.)

205ff   From here onwards the numbering of refs is not in the order of first citation. This relates in particular to refs 34 – 43.

257ff   The significance and interaction of Ran-GTPase and karyopherin in the context should be explained more transparently.

294ff   Discussion. Much of the work discussed from A Palmenberg’s group assesses molecular mechanisms of interaction in great detail; such approach is sadly lacking in this review.

299      Fig. 2. The clarity of Fig. 1 is not higher than that of Fig. 1. Again, molecular mechanisms of enterovirus-induced cleavage events are not discussed.

336      Fig. 3 is not a significant improvement on Figs. 1-2.

349ff   The addition of data of SARS-CoV-2 in the context is insufficiently justified.

358      End of Discussion. The reviewer missed the assessment of earlier data on the interaction of picornavirus proteins with nuclear structures, such as:

Aminev AG, Amineva SP, Palmenberg AC. Encephalomyocarditis virus (EMCV) proteins 2A and 3BCD localize to nuclei and inhibit cellular mRNA transcription but not rRNA transcription. Virus Res. 2003 Sep;95(1-2):59-73.

Aminev AG, Amineva SP, Palmenberg AC. Encephalomyocarditis viral protein 2A localizes to nucleoli and inhibits cap-dependent mRNA translation. Virus Res. 2003 Sep;95(1-2):45-57

Groppo R, Brown BA, Palmenberg AC. Mutational analysis of the EMCV 2A protein identifies a nuclear localization signal and an eIF4E binding site. Virology. 2011 Feb 5;410(1):257-67. Erratum in: Virology. 2011 Dec 5;421(1):85.

Ciomperlik JJ, Basta HA, Palmenberg AC. Three cardiovirus Leader proteins equivalently inhibit four different nucleocytoplasmic trafficking pathways. Virology. 2015 Oct;484:194-202.

Reviewer 2 Report

This in a very informative review describing perturbation of nucleocytoplasmic traffic observed after picornavirus infection.

There are only few points to be addressed in a revised version:

1. lines 62,286, 287: It should read "Aphthovirus"

2. line 65: References 4 and 5 are the original papers of 1988. I would prefer one or two recent reviews here.

3. lines 104-106 and elsewhere in the text: Please explain abbreviations.

4. line 111-112: It should read "rhinovirus 2Apro".

5. lines 172-175: Please, simplify this sentence, e.g. two sentences.

6. lines 257, 259, 261, 350, 356: Do not capitalise "picornavirus" in the main text.

Round 2

Reviewer 1 Report

Nucleocytoplasmic trafficking perturbation induced by picornaviruses

By B Lizcano Perret and T Michiels* (*Corresponding author)

Submitted to Viruses (Editorial No. viruses-1246815 R1)

General Comments

This is the revised version of a manuscript the original submission of which has been studied and commented upon by this reviewer. The authors have considered the suggestions/views of two reviewers very carefully and have thoroughly revised the manuscript which has been much improved. The flow of information between Figs. 1 -3 has been clarified, and the figs have been improved. The addition of refs to Table 2 has made the information more transparent. This reviewer has relatively few additional specific comments.

Specific Comments

Line

13        Consider reading: … transported in both directions.

30        Consider omitting … On the one hand… on the other hand.

47        … Finally, RAN-GDP is returned to the nucleus by…

51ff     Fig. 1, Legend. The legend has much improved. For further improvement of the figure it should be considered to identify the term ‘proteinNLS’ by a symbol in which the protein also has a structure.                                                                                                                  Line 54. Read: … envelope…                                                                                                 References should be provided for the diagrammatic structures shown.

78        Omit ‘already’.

86ff     Reconsider the headings: The crime… the victims… the aggressors… vandalizing… additional victims…

112      Table 1. This reviewer is still not convinced of the value of Table 1. Its content could be replaced by 2 sentences in the Text.

157      Table 2 has much improved by the addition of relevant refs.

204      and 288. This reviewer is not happy about unpublished data being introduced ‘through the backdoor’. Consider to omit this text.

262      … components of the soluble phase… [also in line 304]

341      … STAT proteins…

356ff   If authors insist to take SARS-CoV-2 ‘just as an example’, they should consider the addition of a comment on influenza viruses the nuclear component of the replication cycle of which is very well characterized.

368      This last sentence is incomplete. Consider rephrasing or omitting it.

408      Ref. 14. Its title should not be in capitals.

414      Ref. 17 is incomplete.

521      Ref. 75 is incomplete.

Author Response

General Comments

This is the revised version of a manuscript the original submission of which has been studied and commented upon by this reviewer. The authors have considered the suggestions/views of two reviewers very carefully and have thoroughly revised the manuscript which has been much improved. The flow of information between Figs. 1 -3 has been clarified, and the figs have been improved. The addition of refs to Table 2 has made the information more transparent. This reviewer has relatively few additional specific comments.

> We thank this reviewer for having taken the time to review the second version of the manuscript and for the very careful examination of the text.

Specific Comments

Line

13        Consider reading: … transported in both directions.

> this was modified accordingly

30        Consider omitting … On the one hand… on the other hand.

> this was done

47        … Finally, RAN-GDP is returned to the nucleus by…

> corrected

51ff     Fig. 1, Legend. The legend has much improved. For further improvement of the figure it should be considered to identify the term ‘proteinNLS’ by a symbol in which the protein also has a structure.                                                                                                           

> Fig 1 has been modified to enlarge the protein, which contains the NLS to make it clear that the NLS is part of the protein. The protein has been renamed "cargo" to fit the text better.

            Line 54. Read: … envelope… 

> thanks... this was corrected                                                                                              

References should be provided for the diagrammatic structures shown.

> references were added although they were already provided in the text.

78        Omit ‘already’.

> OK, this was done

86ff     Reconsider the headings: The crime… the victims… the aggressors… vandalizing… additional victims…

> These headings were modified to more neutral ones

112      Table 1. This reviewer is still not convinced of the value of Table 1. Its content could be replaced by 2 sentences in the Text.

> Table 1 was suppressed, as recommended.

157      Table 2 has much improved by the addition of relevant refs.

> Thank you

204      and 288. This reviewer is not happy about unpublished data being introduced ‘through the backdoor’. Consider to omit this text.

> The text that was added line 288, during the first revision process, was deleted. We wish, however, to keep the unpublished data provided in Table 1 including those kindly communicated by M. Saeed since they are very pertinent to this review.

262      … components of the soluble phase… [also in line 304]

> This was corrected

341      … STAT proteins…

> corrected

356ff   If authors insist to take SARS-CoV-2 ‘just as an example’, they should consider the addition of a comment on influenza viruses the nuclear component of the replication cycle of which is very well characterized.

> We added references to influenza as well as to additional pathogens, such as Zika virus and even bacteria, to end the discussion with a broader opening on the topic.

368      This last sentence is incomplete. Consider rephrasing or omitting it.

> this was modified

408      Ref. 14. Its title should not be in capitals.

414      Ref. 17 is incomplete.

521      Ref. 75 is incomplete.

> Thank you; References were updated manually as these errors apparently stemmed from the bibliography software that was used.